# Amphiphilic Comb Polymers as New Additives in Bicontinuous Microemulsions

**DOI:** 10.3390/nano10122410

**Published:** 2020-12-02

**Authors:** Debasish Saha, Karthik R. Peddireddy, Jürgen Allgaier, Wei Zhang, Simona Maccarrone, Henrich Frielinghaus, Dieter Richter

**Affiliations:** 1Solid State Physics Division, Bhabha Atomic Research Centre, Mumbai 400085, India; debiitm@gmail.com; 2Department of Physics and Biophysics, University of San Diego, San Diego, CA 92110, USA; kpeddireddy@sandiego.edu; 3Jülich Centre for Neutron Science (JCNS-1) and Institute of Biological Information Processing (IBI-8) Forschungszentrum Jülich GmbH, 52425 Jülich, Germany; j.allgaier@fz-juelich.de (J.A.); w.zhang@fz-juelich.de (W.Z.); d.richter@fz-juelich.de (D.R.); 4Jülich Centre for Neutron Science (JCNS), Forschungszentrum Jülich GmbH, Outstation at FRM II, Lichtenbergstr. 1, 85747 Garching, Germany; s.maccarrone@fz-juelich.de

**Keywords:** amphiphilic polymer, bicontinuous microemulsion, small-angle neutron scattering, phase diagram, efficiency increase

## Abstract

It has been shown that the thermodynamics of bicontinuous microemulsions can be tailored via the addition of various different amphiphilic polymers. In this manuscript, we now focus on comb-type polymers consisting of hydrophobic backbones and hydrophilic side chains. The distinct philicity of the backbone and side chains leads to a well-defined segregation into the oil and water domains respectively, as confirmed by contrast variation small-angle neutron scattering experiments. This polymer–microemulsion structure leads to well-described conformational entropies of the polymer fragments (backbone and side chains) that exert pressure on the membrane, which influences the thermodynamics of the overall microemulsion. In the context of the different polymer architectures that have been studied by our group with regards to their phase diagrams and small-angle neutron scattering, the microemulsion thermodynamics of comb polymers can be described in terms of a superposition of the backbone and side chain fragments. The denser or longer the side chain, the stronger the grafting and the more visible the brush effect of the side chains becomes. Possible applications of the comb polymers as switchable additives are discussed. Finally, a balanced philicity of polymers also motivates transmembrane migration in biological systems of the polymers themselves or of polymer–DNA complexes.

## 1. Introduction

The first promising polymers that were added to microemulsions were amphiphilic diblock copolymers [1,2,3], and they remain a topic of active research [4]. The required amount of surfactant could be reduced by this additive, resulting in formulations with a reduced environmental impact (polymer-boosting effect). Over the years, many different amphiphilic polymer architectures have been examined [5,6,7,8,9] with differing effects on the microemulsion. One area of polymer architecture that has yet to be explored is that of amphiphilic comb polymers. These would appear to be particularly promising as they can be cheaply synthesized using grafting reactions and have already found uses as gene carriers and scaffolds for tissue engineering [10,11,12].

Microemulsions [13] consist of oil, water, and a surfactant that mediates between the two, otherwise immiscible, components. Microemulsions form spontaneously and thus are thermodynamically stable. On the nanoscale, the surfactant forms a film between the two immiscible domains. The phase diagram of microemulsions [14] displays a wide variety of different domain structures that are identified by various methods [15,16]. If equal amounts of oil and water are used, a bicontinuous phase may form a sponge-like structure, where the oil interpenetrates the water and vice versa. The minimum amount of surfactant needed to find this phase is related to the efficiency of the surfactant.

Amphiphilic diblock copolymers are found to be anchored in the membrane with each of the two blocks placed in the domain of preferred solubility [1]. The configurational entropy, i.e., the thermal movement of each block, exerts an effective pressure on the surfactant membrane that acts to reduce the local curvature [17,18,19,20]. In this way, the surface area to volume ratio is minimized, thereby reducing the amount of surfactant required, i.e., increasing the surfactant efficiency. Amphiphilic sticker polymers [5] are highly asymmetric with only one polymeric block, but still have the ability to anchor in the surfactant membrane. As a result, the efficiency is increased here, too. However, the asymmetry also increases the membrane curvature. In telechelic polymers, both ends of the polymer chain are able to anchor in the surfactant membrane [6,21]. Here, the domain size determines if the anchoring points are close together (large domains) or if the polymer bridges two oil domains. While the first effect leads to reduced efficiency, the latter enhances the efficiency. This makes the polymer suitable for switching applications, whereby the polymer can either support a desired phase separation or enhance the efficiency depending on an external stimulus. The domain size effects have also been investigated in the case of diblock copolymers [8]. The confinement of the polymer by the membrane pushes the polymer against the membrane and, therefore, the pressure on the membrane increases, which leads to increased efficiencies. Apart from the encapsulation of telechelic and diblock copolymers at intermediate confinement as discussed so far, the confinement can be even stronger [6,8]. In this case, the polymer conformation bounces strongly between membrane walls, and the architecture of the polymer (telechelic, diblock, or homopolymer), i.e., the anchoring no longer plays a role. In such cases, all polymers display substantially reduced efficiencies (antiboosting) as for normal homopolymers [6,8]. The degree of confinement for all polymers is described by the ratio of the domain size, d, and the polymer end-to-end distance, *R*_ee_.

The amphiphilic comb polymers of this study consist of an oil-soluble poly(1,2-butylene oxide) (PBO) backbone and water-soluble poly(ethylene oxide) (PEO) side chains. If the concepts learned so far can be applied to this type of polymer, the side chains would also anchor in the membrane and stay in the water domain, and the backbone will form loops in the oil domain between the anchoring points (apart from the first and last fragments that may act as sticker polymers). In this sense, the polymer is fragmented by the anchoring. This raises the questions as to whether we can describe the polymer effect on the efficiency by a linear superposition of all fragments independently and to what extent can we neglect cross-correlations between the different fragments?

For completeness, we also studied the homopolymer polyethylene oxide (PEO or PEG) with the ionic surfactant dioctyl sulfosuccinate sodium (AOT). From the literature [22,23,24,25], it is known that the polymer also anchors at the membrane due to the interaction between the oxygen atoms of the PEG with the sodium ions without the presence of side chains. This sample with a side chain length of zero, therefore serves a point of comparison for the comb polymers. Finally, a summary of all polymer characteristics (molar masses of backbone fragment and side chain) is displayed in Figure 1 for all polymers featured in this investigation. For the linear superposition assumption of all fragments, the molar mass of the whole polymer does not play a role.

The manuscript is organized as follows: we first describe the materials and methods applied (with their connection to theory), including the contrast variation method and the scattering models of the anchored polymer. All of this information is based on an extensive, previously published work [1] that we only briefly introduce. We then discuss the effect of the polymer additive on the microemulsion characteristics, confirm the segmented polymer structure in the membrane using contrast variation SANS measurements, and present one extensive phase diagram study for one comb polymer. In the discussion, we confirm the superposition of all polymer fragments for the coefficients of the microemulsion characterization. By increasing the density or length of the side chains, brush formation is enhanced, which, in turn, increases the effect of the polymers of the membrane. In the summary, the role of the experiments and comb polymers, in general, is reviewed.

## 2. Materials and Methods

### 2.1. Polymer Synthesis and Characterization

The amphiphilic comb polymers used for this study contain hydrophobic PBO backbones and hydrophilic PEO side chains. PBO with a narrow molecular weight distribution was obtained by anionic ring opening polymerization of BO monomer at low temperatures in the presence of crown ethers [26]. In this case, BO was copolymerized with 1,2-epoxy-7-octene (EOc) in order to functionalize the backbone with vinyl groups [27]. In the next step, 9-BBN/H_2_O_2_ was used to oxidize the vinyl groups to primary alcohols, which, in turn, were used as starting points for the polymerization of the PEO side chains.

Under the polymerization conditions chosen, the EOc units are incorporated in the backbone in a fully random fashion. This was proven by analyzing aliquots taken during the polymerization reactions of some of the backbone syntheses. In all the cases, the monomer composition stayed constant during the reactions and was identical with the originally used monomer mixture within the experimental error. The backbone polymerizations were terminated by methylation of the alkanolate chain end with CH_3_I. This measure prevented the subsequent creation of additional PEO blocks at one backbone chain end. Table 1 shows the narrow molecular weight distributions of the backbones.

In the following reaction step, the backbone vinyl groups were first hydroborated with 9-BBN, followed by oxidation with H_2_O_2_/NaOH. As a result, selectively terminal alcohol groups were created. In the final step, the alcohol groups of the backbone were used as initiating units for the side chain polymerization. First, the alcohol groups were partially metalated using potassium. Knowledge of the alcohol group concentration from ^1^H NMR allowed for a precise adjustment of the metalation degree. The PEO side chains were polymerized by adding EO monomer. In order to obtain combs with different side chain numbers, the fraction of oxidized vinyl groups was varied. Appendix A shows the SEC traces of the nonfunctionalized backbone and the final comb product for Comb 8. From the chromatogram of the product, it is visible that no backbone precursor remained in the product. This is also the case for the combs with lower side chain numbers. The small shoulder in the comb signal at lower retention times is not present in the nonfunctionalized backbone. It appeared after the oxidation step and is probably attributable to some cross-linking reaction of the boronic ester intermediate. The characterization results for the combs listed in Table 1 confirm the narrow molecular weight distributions of all comb polymers. The comb synthesis is outlined in more detail in the Appendix A, together with the description of the characterization procedures.

### 2.2. Further Materials

The synthesis of the deuterated nonionic surfactant d-C_10_E_4_ has been described in Ref. [1]. The same batch of material was also used in this study. The materials PEG (4.6 k), AOT surfactant, and deuterated THF were purchased from Sigma Aldrich (Taufkirchen, Germany). Deuterated (heavy) water and deuterated n-decane were purchased from Armar Chemicals, Döttingen, Switzerland. The hydrogenous nonionic surfactant, h-C_10_E_4_, was purchased from Bachem Chemicals, Weil am Rhein, Germany. All chemicals were used as received. Deionized water was obtained from a PURELAB Ultra filter from ELGA (Berlin, Germany) at 18.2 MΩcm.

### 2.3. Phase Diagram Measurements

The diversity of measurements to determine phase diagrams of microemulsions is described in the literature [15,16]. The most important property for distinguishing a single-phase bicontinuous microemulsion from a phase-separated system is the turbidity. A one-phase system appears transparent, while a phase-separated system appears turbid. For nonionic surfactant microemulsions, the essential parameters are the temperature, *T*, and the membrane volume fraction, γ, (simply called surfactant content) with the otherwise symmetric microemulsion:(1)$$\gamma =\frac{m_{s}/\rho_{s}-0.02\cdot m_{o}/\rho_{o}}{m_{o}/\rho_{o}+m_{w}/\rho_{w}+m_{s}/\rho_{s}+m_{p}/\rho_{p}}$$
with the masses, *m_i_*, and densities, $\rho_{i}$, of the specific components (index *i* refers to the material: o = oil, w = water, s = surfactant, p = polymer). For the nonionic surfactant, a 2% solubility of unimer [28] in oil is considered (this can be neglected in the case of AOT). Contrary to the first publication [2] where the total amphiphile concentration (consisting of surfactant and polymer) was considered to be essential for the system, we subsequently found [5,6,8] that the membrane volume fraction is only marginally altered by the anchoring of the polymers that are mainly found to be dissolved in the oil and water domains. However, we define the polymer mass fraction in the total amphiphile content as:(2)$$\delta =\frac{m_{p}}{m_{p}+m_{s}}$$

In addition to the simple visual inspection of the samples to determine the turbidity, we employed crossed polarizers to observe the lamellar liquid crystalline phase at higher surfactant concentrations. For the polymer, Comb 3, even more liquid crystalline phases (as one-phase regions and as coexisting phases) were observed in an extensive study. After complete phase separation, the number of coexisting phases could be determined by the number of meniscuses. From small-angle neutron scattering and crossed polarizer experiments on single phases, the exact state or structure of the phases could be determined.

### 2.4. Small-Angle Neutron Scattering (SANS)

SANS measurements were conducted at the KWS1 instrument [29,30] at the research reactor FRM-2, hosted by the Heinz Maier-Leibnitz Zentrum (MLZ) (Garching, German) and operated by the Jülich Center for Neutron Science (JCNS). We set the wavelength to λ = 4.5 Å and had the following collimation with respect to detector distance in units of meters: 20-8 (i.e., a collimation length of 20 m and a sample-to-detector length of 8 m) and 20-2 for the bulk contrast of the microemulsions and 20-20, 8-8, and 8-2 for the contrast variation experiments. The measurements were corrected for background and dark current and calibrated using a secondary standard of plexiglass. The scattering vector was calculated on the basis of the scattering geometry: $Q=4\pi /\lambda \;\sin\left(\theta /2\right)$ with θ being the scattering angle. The data were averaged azimuthally to obtain 1-dimensional data sets, i.e., the macroscopic cross-section as a function of *Q* on absolute scales: $d\Sigma /d\Omega \left(Q\right)$.

## 3. Analysis Methods

### 3.1. Theory of Microemulsion Characterization

The characterization of bicontinuous microemulsions with polymers has been standardized by our group to a large extent [1,5,6,7,8,9]. For this, we employed phase diagram measurements [15,16] and small-angle neutron scattering on the domain structure (bulk contrast) as a function of polymer content. By employing the theoretical framework as developed in Refs. [31,32], we could extract the variation of the saddle splay modulus and the bending rigidity with the polymer content that could then be compared with the theory. We first aim to determine the coefficients of these two dependencies and will subsequently compare them with experimental values of diblock copolymers [8] and telechelic polymers [6] since those aspects also appear with the polymer backbone and side chains.

### 3.2. Phase Diagrams

The principal phase diagram of a symmetric microemulsion with equal volumes of oil and water displays the temperature, *T*, as a function of the surfactant content, γ, (see Figure 2). At very high temperatures, water is expelled due to the strong curvature of the surfactant film towards the water, and a two-phase system, $\bar{2}$, is obtained. Equally, the oil is expelled at low temperatures, where the two-phase system, 2, is found. At an intermediate temperature, the curvature is balanced and domain structures of minimal surfaces are observed. At high surfactant concentrations, the full amount of oil and water can be dispersed, and a one phase bicontinuous microemulsion, 1, is observed. This phase is in the focus of our studies. Conversely, at low surfactant content, the bicontinuous microemulsions expel both oil and water, and so a three-phase system, 3, is obtained. At very high surfactant contents, the microemulsion might even take on a lamellar structure, but for simplicity, we do not consider this behavior. The critical point of stability, $\tilde{X}$, is given by the minimum amount of surfactant, $\tilde{\gamma }$, required for the one-phase system and the phase inversion temperature, $\tilde{T}$. The resemblance of the phase diagram to a fish leads to the term “fish tail point” for $\tilde{X}$.

.

Variations of the negative intrinsic saddle splay modulus, ${\bar{\kappa }}_{0}$, are connected to the minimum amount of surfactant, $\tilde{\gamma }$, according to [31]:(3)$$\Delta {\bar{\kappa }}_{0}=-\frac{\bar{\alpha }}{4\pi }\cdot \Delta \ln\left(\tilde{\gamma }\right)=-\hat{\Xi }\cdot \Delta \sigma \cdot R_{\mathrm{ee}}^{2}$$
where $\bar{\alpha }=-10/3$, $\hat{\Xi }$ is the sensitivity of the polymer effect, σ is the polymer number density on the surfactant film, and *R*_ee_ is the end-to-end distance of the polymer. All delta values should be interpreted in the context of continuous addition of the same polymer, i.e., the minimum surfactant amount, $\tilde{\gamma }$, varies with the polymer density, σ. Originally, this formula was derived for diblock copolymers with a single anchoring point in the membrane. For comb polymers, the different polymer parts of side chains and backbone fragments would lead to different interpretations of the term $R_{\mathrm{ee}}^{2}$ (Figure 3). Therefore, the total effective end-to-end distance term is calculated according to:(4)$$R_{\mathrm{ee}}^{2}=N_{\mathrm{SA}}R_{\mathrm{eeSC}}^{2}+\frac{2}{2}N_{\mathrm{BB}}R_{\mathrm{eeBB}}^{2}$$

The number of side chains is *N*_SC_ and the number of backbone fragments is *N*_BB_, while the end-to-end distances of the side chains and backbone fragments are *R*_eeSC_ and *R*_eeBB_, respectively. A sketch of this interpretation is found in Figure 3, where the segmentation is indicated. The side chains are taken as full segments, while the backbone fragments are interpreted as two independent fragments of half the length. For the PEO side chains in water, we calculate the end-to-end distance according to [33]:(5)$$R_{\mathrm{eeSC}}^{2}=6\cdot 0.0408\;{Å}^{2}\cdot {\left(M_{\mathrm{SC}}\left[g/\mathrm{mol}\right]\right)}^{1.16}$$

The molar mass of the side chain is *M*_SC_. For the whole PBO backbone fragments in n-decane, we assume a simple scaling with the monomeric molar masses (44/72) compared to the PEO monomer:(6)$$R_{\mathrm{eeBB}}^{2}=6\cdot 0.0408\;{Å}^{2}\cdot {\left(\frac{44}{72}M_{\mathrm{BB}}\;\left[g/\mathrm{mol}\right]\right)}^{1.16}$$

The molar mass of the backbone fragment is *M*_BB_. From the scaling of the end-to-end distances with the experimentally determined exponent [33] of 1.16 in a good solvent (Equations (5) and (6)) and the total contribution to the polymer size (Equation (4)), we can approximately state that the details of segmentation are of limited importance in this case (because 1.16 is close to unity). The polymer number density at the surfactant membrane is calculated by:(7)$$\sigma =\frac{m_{p}M_{s}}{m_{s}M_{p}}\cdot A_{s}^{-1}$$
with the molecular masses *M*_s_ = 334.5 g/mol and *M*_p_ of the surfactant and the whole polymer and the masses *m*_s_ and *m*_p_ of surfactant and polymer present in the sample. The area per C_10_E_4_ surfactant molecule is *A*_s_ = 54 Å^2^ [34,35]. The relationship between the amphiphilic weights *m*_s_ and *m*_p_ and the polymer ratio, δ, is:(8)$$\frac{m_{p}}{m_{s}}=\frac{\delta }{1-\delta }$$

A further property that can be extracted from the phase diagram is the mean curvature, 〈*H*〉, as function of the temperature change, ΔT, given by:(9)$$\langle H\rangle =\mu \cdot \Delta T=ϒ\Delta \sigma \cdot R_{\mathrm{ee}}^{1}$$

The experimental coefficient, *μ*, is −1.4 × 10^−3^ Å^−1^ K^−1^ for the surfactant C_10_E_4_ [35] and estimated to be −1.5 × 10^−4^ Å^−1^ K^−1^ for AOT [36]. The coefficient ϒ, is the sensitivity of the mean curvature to the polymer. Due to the different scaling of the end-to-end distance, *R*_ee_, the following simple approach is elaborated:(10)$$\langle H\rangle =ϒ\cdot \Delta \sigma \cdot \left(N_{SA}R_{\mathrm{eeSC}}^{1}+\frac{2}{\sqrt{2}}N_{BB}R_{\mathrm{eeBB}}^{1}\right)$$

Because telechelic polymers drag the membrane curvature towards the polymer (or here oil) side [6] and the side chains [5] push the membrane away towards the oil domain here, the two polymer effects are added up in Equation (10). Due to mismatching exponents in Equation (10), the polymer segmentation does matter. To our belief, this is the most reasonable segmentation.

### 3.3. Small-Angle Neutron Scattering on Domains

When rendering the large contrast between the oil and water domains (usually by using heavy water), the dominant scattering results from the oil and water domains and the corresponding contrast is called bulk contrast. The scattering of the added polymer itself can be safely neglected, and so we only observe the influence of the polymer on the domain structure. From diblock copolymers experiments [1], we learned that the configurations of an anchored polymer exert a pressure on the membrane that reduces the membrane fluctuations and a microemulsion with a lower surface area to volume ratio is formed. The membrane fluctuations are a measure for the bending rigidity, κ, and the small angle scattering is connected to κ via the Gaussian random field theory [32].

The bulk scattering of a bicontinuous microemulsion was first described by Teubner and Strey [37] who argued, on the basis of the Landau free energy and the system symmetry, that the small angle scattering has a typical correlation peak. The Landau approach comes from the long wavelength limit and does not cover all local fluctuations of the membrane. We have therefore included an additional term [38] that accounts for the correct Porod constant at larger *Q* with the correct surface area to volume ratio. The exponent 4 accounts for the sharp domain interfaces. So, we arrive at the following small angle scattering:(11)$$\frac{d\Sigma }{d\Omega }\left(Q\right)=\left(\frac{A}{{\left(k_{0}^{2}+\xi^{-2}\right)}^{2}-2\left(k_{0}^{2}-\xi^{-2}\right)Q^{2}+Q^{4}}+B{\left[\frac{{\mathrm{erf}}^{3}\left(1.06\;QR_{g}/\sqrt{6}\right)}{Q}\right]}^{4}\right)\exp\left(-\sigma_{r}^{2}Q^{2}\right)+b_{\mathrm{ckgr}}$$

The coefficient $A=8\pi \langle \nu^{2}\rangle /\xi$ is connected to the domain fluctuations measured by ν. The structural parameters of the Teubner–Strey model are the domain repeat distance, *d*_TS_, (with $k_{0}=2\pi /d_{\mathrm{TS}}$) and the correlation length, ξ. The sum of the two coefficients, *A* and *B*, yields the Porod constant *P* = *A* + *B*. The term in the square brackets was taken from the Beaucage model [39] of fractal structures as a term describing the scattering at larger angles, i.e., larger *Q* values. The first two terms are multiplied by a Gaussian with the surfactant film roughness $\sigma_{r}$ arising from a smearing due to thermal fluctuations of single molecules in the surfactant film. The incoherent background is denoted *b*_ckgr_. The structural parameters of the Teubner–Strey model are connected to the bending rigidity via the Gaussian random field model according to [1,32]:(12)$$\frac{\kappa_{\mathrm{SANS}}}{k_{B}T}=\frac{5\sqrt{3}}{64}\cdot k_{0}\xi$$
with the Boltzmann constant, *k*_B_, and the absolute temperature, *T*. The experimental bending rigidity, $\kappa_{\mathrm{SANS}},$ was shown to be a superposition of the bare bending rigidity, κ, obtained from neutron spin echo spectroscopy experiments and the saddle splay modulus, $\bar{\kappa }$, obtained from phase diagram measurements [40]. Since the bare bending rigidity only makes small a small contribution to $\kappa_{\mathrm{SANS}}$ (approximately 15%), the extraction of the bare values from SANS measurements result in rather large statistical uncertainties. Thus, throughout this paper, we stick to the experimental bending rigidity, $\kappa_{\mathrm{SANS}},$ which we compare with results from other polymers [6,8]. For the dependence of the bending rigidity on the polymer content, we obtain a similar result to that obtained in Equation (3) [32]:(13)$$\Delta \kappa_{\mathrm{SANS}}=\Xi_{\mathrm{SANS}}\cdot \Delta \sigma \cdot R_{\mathrm{ee}}^{2}$$

Again, as for the saddle splay modulus (Equation (3)), we find that the segmentation does not greatly influence the bending rigidity. The three polymer specific coefficients, i.e., $\Xi_{\mathrm{SANS}}$, $\hat{\Xi }$, and ϒ, will then be compared with results on other polymers such as diblock copolymers, telechelic polymers, and sticker polymers. These comparisons will reveal that the mechanisms by which the comb polymer acts on the membrane is equivalent to the presence of many single comb polymer fragments as discussed in Figure 3.

### 3.4. Contrast Variation SANS Measurements and Singular Value Decomposition

The observed scattering intensity, $d\Sigma /d\Omega \left(Q\right),$ is connected to the set scattering length densities, $\rho_{i}$, and the partial scattering functions, $S_{ij}$(*Q*), according to [1]:(14)$$\frac{d\Sigma }{d\Omega }\left(Q\right)=\underset{i,j}{\sum }\left(\rho_{i}-\rho_{w}\right)\left(\rho_{j}-\rho_{w}\right)S_{i,j}\left(Q\right)$$
where the indices *i* and *j* represent oil (o), film or surfactant (f), or polymer (p) and the water parameters (w) are mentioned explicitly. Even though there are two indices $\left\{i,j\right\}=:k$, a unified index k can describe all 6 different contrast conditions. Usually, the experimental conditions of contrasts indexed by ℓ are many more than the 6 different principal contrasts, and so the partial scattering functions are overdetermined (see Appendix A). The matrix of the different experimental contrasts can be formally inverted by the singular value decomposition method, and so, the partial scattering functions are determined with a high degree of experimental accuracy:(15)$$S_{k}\left(Q\right)={\left(\Delta \rho_{k,\ell }^{2}\right)}^{-1}{\left.\frac{d\Sigma }{d\Omega }\right|}_{\ell }\left(Q\right)$$

Details of this method are described elsewhere [1]. We call the self-correlations *S*_oo_ bulk contrast, *S*_ff_ film contrast, and *S*_pp_ polymer contrast. From the polymer contrast, many details of the polymer structure can be deduced. In theory, the cross terms *S*_op_, *S*_fp_, and *S*_of_ provide information on the relative placement of two respective components. Within our modeling abilities and also due to the reduced statistical confidence, we largely neglect the cross terms and focus on the polymer scattering, *S*_pp_.

### 3.5. Polymer Scattering Model

Here, we describe a method for modeling the polymer scattering in two different solvent conditions: in a moderately good or theta solvent (see supporting information) and at the membrane with the backbone and side chains dissolved in the different oil and water domains, respectively. The modeling of the polymer in a theta solvent means that both backbone and side chain monomers are energetically neutral when in contact with the solvent and with each other. The elementary function for simple homopolymers was derived by Debye [41]. The crux of this calculation is that the scattering of two monomers is essentially determined by the number of monomers between them [42,43], along the polymer structure. In a comb polymer, this number of monomers is well defined. The detailed formulae are given in the supporting information.

For modeling the polymer at the surfactant membrane, we assume that the membrane is locally flat (for simplicity). Furthermore, we assume that the backbone fractions are divided into two equal parts, each of them being closest to one side chain (see Figure 3). This separates the polymer into Y-shaped polymer fragments with the side chain in the aqueous domain and the two backbone fragments in the oil domain. The ends of the side chains are able to move freely, and we have assumed that the same is also true of the backbone fragments in the direction normal to the membrane. From these assumptions, we obtain the analytic expression:(16)$$R_{z}\left(Q_{z}\right)=4{\left|\left(\Delta \rho_{S}V_{S}\right)\frac{R\left(Q_{z}R_{\mathrm{gS}}\right)-R\left(\frac{1}{2}Q_{z}R_{\mathrm{gS}}\right)}{Q_{z}R_{\mathrm{gS}}}+\left(\Delta \rho_{B}V_{B}\right)\frac{R\left(Q_{z}R_{\mathrm{gB}}\right)-R\left(\frac{1}{2}Q_{z}R_{\mathrm{gB}}\right)}{Q_{z}R_{\mathrm{gB}}}\right|}^{2}$$
where the complex Dawson function is defined as:(17)$$R\left(x\right)=\exp\left(-x^{2}\right)\left(-i+\overset{x}{\underset{0}{\int }}dt\exp\left(t^{2}\right)\right)$$

In the normal direction to the membrane, the connectivity of the backbone is neglected, which should be a good approximation for long backbones. Unfortunately, there is no better approximation at hand. For the lateral structure, we focus on the stretched backbone fragments. They are stretched due to the repulsive interactions of the side chains on the other side of the membrane. For a random walk with a trend, $\vec{R}$, and a radius of gyration, *R*_g_, we derived the following formula (diffusion in a constant, homogenous flow):(18)$$\begin{array}{cc} P_{\text{mod-Debye}}(\vec{Q})= & \frac{2}{{\left(Q^{2}R_{g}^{2}+{\left(\vec{Q}\vec{R}\right)}^{2}\right)}^{2}}\{[\cos(\vec{Q}\vec{R})\left(Q^{2}R_{g}^{2}+{\left(\vec{Q}\vec{R}\right)}^{2}\right)\\ & -2\sin(\vec{Q}\vec{R})\left(Q^{2}R_{g}^{2}\right)(\vec{Q}\vec{R})]\exp\left(-Q^{2}R_{g}^{2}\right)-{\left(Q^{2}R_{g}^{2}\right)}^{2}+{\left(\vec{Q}\vec{R}\right)}^{2}+{\left(Q^{2}R_{g}^{2}\right)}^{3}\\ & +\left(Q^{2}R_{g}^{2}\right){\left(\vec{Q}\vec{R}\right)}^{2}\} \end{array}$$

We define the number of Kuhn segments of length $\ell_{K}$ within the backbone fragments and single side chains as *N*_KBB_ and *N*_KSC_, respectively, while the number of each fragment type is *N*_BB_ and *N*_SC_ (consistent with Equations (5), (6), and (21)). We count backbone fragments on either side of the first and last side chains, thus *N*_SC_ = *N*_BB_-1, and (*N*_BB_-2) is the number of backbone fragments in between side chains. We obtain for the total trend of the random walk of the polymer:(19)$$\left|\vec{R}\right|=\left(N_{\mathrm{BB}}-2\right)N_{\mathrm{KBB}}\alpha \ell_{K}$$
with the fraction of directed monomers, α. The fraction of random polymer steps is given by $\left(1-\alpha \right)$ and so in 2 dimensions, the radius of gyration is:(20)$$R_{g}^{2}=\frac{1}{4}\left(N_{\mathrm{BB}}-2\right)N_{\mathrm{KBB}}\left(1-\alpha \right)\ell_{K}^{2}$$

Within the Kuhn model (we also refer to Equations (5) and (6)), the following relations hold:(21)$$R_{\mathrm{eeBB}}^{2}=N_{\mathrm{KBB}}\ell_{K}^{2}\;\mathrm{and}\;\ell_{K}=C_{\infty }\ell \;\mathrm{and}\;N_{\mathrm{KBB}}=N/C_{\infty }$$
with the counting of chemical bonds, *N*, along the chain and an assumed average bond length ℓ = 1.54 Å within a backbone fragment. For the scattering from the backbone as a whole, we assume a semiflexible chain model:(22)$$P\left(Q\right)=\frac{2}{Q^{4}R_{g2}^{4}}(\exp(-Q^{2}R_{g2}^{2})-1+Q^{2}R_{g2}^{2})+\frac{1}{N_{2}}\left({\langle P_{\mathrm{mod}-\mathrm{Debye}}\rangle }_{\varphi }-\frac{1}{Q^{2}R_{g2}^{2}}(1-\exp(-Q^{2}R_{g2}^{2}))\right)$$
where the approximate number of stretched Kuhn fragments in the overall backbone is given by:(23)$$N_{2}=\frac{1}{2}N_{\mathrm{SC}}/\left(N_{\mathrm{BB}}-2\right)$$
and the overall radius of gyration is:(24)$$R_{g2}^{2}=N_{2}{\left|\vec{R}\right|}^{2}$$
which are derived in the sense of persistence fragments of length $\left|\vec{R}\right|$. The modified Debye scattering ${\langle P_{\text{mod-Debye}}\rangle }_{\varphi }$ is orientationally averaged in the x-y-plane. The total polymer scattering is then:(25)$$\frac{d\Sigma }{d\Omega }\left(Q\right)=\langle \frac{\phi_{\mathrm{polymer}}}{V_{\mathrm{polymer}}}\cdot R_{z}\left(Q_{z}\right)\cdot P{\left(Q_{xy}\right)\rangle }_{\vartheta }$$
which is orientationally averaged over the azimuthal angle, ϑ. The polymer scattering was treated on an absolute scale to account for possible differences in the monomeric scattering length densities $\Delta \varrho_{S}$ and $\Delta \varrho_{B}$. To compare this with the deconvoluted scattering functions, which are not contrast dependent, we simply divided by the average contrast. Within the formula of Equation (25) we neglected the side chain scattering in the x-y-plane that would result from very short (Comb 4) chain scattering. A comparison to the experimental data supports this omission, as the contribution from short length scales was found to be negligible.

## 4. Results

The experimental phase diagram for bicontinuous microemulsions of the polymer Comb 4 is depicted in Figure 4. We can identify that the one-phase region, i.e., the fish tail, is shifted towards higher surfactant concentrations, γ, in the presence of Comb 4 compared to the system with no additive. Increasing the polymer content also increases the amount of surfactant required to produce a one-phase microemulsion, i.e., the efficiency is reduced. Increasing the polymer content also has the effect of increasing the phase inversion temperature. Taking the logarithm of the minimum surfactant amount as a function of the scaled polymer density (Figure 5), we obtain the coefficient $\hat{\Xi }$ that evaluates to −0.528 (Equation (3)). Similarly, the dependence of the phase inversion temperature of scaled polymer density (particular for the membrane curvature, see Figure 6) leads to the coefficient ϒ, which evaluates to −0.202 (Equation (9)).

The SANS curves from Comb 4 are depicted in Figure 7. From these measurements, we can deduce the apparent bending rigidity, $\kappa_{\mathrm{SANS}}$ (Figure 8a). The large amounts of data reduce to single values for each membrane content by interpolation to γ = 0.19. A single function of $\kappa_{\mathrm{SANS}}$ dependent on the scaled polymer content is obtained (Figure 8b). A linear dependence within the error bars is observed for the first three data points, which compares well in terms of data points with the other comb polymers. We obtain the coefficient $\Xi_{\mathrm{SANS}}$ that evaluates to −0.391. The residual data points display a stronger dependence, which hints towards the presence of confinement effects—either along the membrane or between neighboring membranes. The whole analysis of phase diagrams and SANS curves is repeated for all comb polymers and the PEG polymer with AOT as surfactant. Details about the experimental analysis can be found in the Appendix A. Both the magnitude as well as the sign of the coefficients $\hat{\Xi }$, ϒ, and $\Xi_{\mathrm{SANS}}$ vary from polymer to polymer.

### 4.1. Contrast Variation Experiments

For the Comb 4 polymer, we conducted contrast variation experiments [1]. A map of the measured contrasts close to the polymer contrast is shown in Appendix A, apart from the bulk and film contrasts. The characteristic scattering functions of the self-correlations is depicted in Figure 9a. While the oil–oil and the film–film correlations correspond extremely well to the bulk and film contrast measurements, the polymer–polymer correlation data provide valuable information for determining how the comb polymer is anchored in the membrane. The cross-correlations are depicted in Figure 9b.

The polymer–polymer correlation was fitted with Equations (16)–(25), where the molar masses of the backbone and side chains were set according to the polymer architecture. The number *N*_stiff_:= *N*_BB_-2 of 16.1 was obtained that is considerably shorter than the number of side chains of 36.9. So, the polymer contains fewer rigid fragments than estimated from the theory in Section 3. A stretching factor α of 0.54 was obtained that is lower than calculated theoretically from the chemistry (i.e., 1.22, see Table 2). So, the lower number *N*_stiff_ seems to compensate the lower stretching α. The value obtained for $C_{\infty }$ of 9.2 is slightly higher than for poly(alkylene oxides) in solution (5.5), as calculated either from Equations (5)/(6) and (21) or from Ref. [44]. From this polymer structure analysis, we have demonstrated that the polymer structure can be described by the model of stretched backbones and side chains dangling in the water. The overall stretching agrees with our theoretical considerations below, but the lower *N*_stiff_ and a slightly elevated $C_{\infty }$ seems to compensate the lower stretching α.

The oil–polymer cross-correlation (Figure 9b) is numerically rather unstable at low *Q* < 0.004 Å^−1^. The Guinier scattering persists until *Q* < 0.02 Å^−1^ and agrees with the Guinier scattering of the polymer–polymer correlation. The sudden decay to negative values relates to the membrane patch size within the bicontinuous microemulsion with a size of approximately 100 Å radius (within this persistence range, the membrane appears flat). The film–polymer correlation (Figure 9b) exhibits a peak at *Q* = 0.05 Å^−1^, which indicates a distance of 125 Å between neighboring film/polymer compartments. This distance does not fit exactly to the half of the film repetition period (approximately 180 Å). So, the presence of polymers distorts the equilibrium film distance. A similar contrast variation analysis on Comb 6 is shown in the supporting information (Appendix A).

The polymer structure in d-THF, a good solvent, is also compared to the theory of Equations (S1)–(S6) in the Appendix A in Figure 9c. Here, all polymer fragments are equally well solvated, and so the structure of the polymer is rather coiled than stretched. This is why the structure in THF appears considerably smaller than that at the membrane. The fitted value for $C_{\infty }$ was 4.7, that agrees well with the value of 5.5 found for poly(alkylene oxides) [44].

### 4.2. Extensive Phase Diagram Measurements

The phase diagram measurements were extended to larger temperature ranges and surfactant amounts for the Comb 3 (Figure 10). It can be seen that the classical fish part of the phase diagram is shifted to higher temperatures. This means that the polymer counteracts against the effects of the elevated temperatures, where the membrane naturally curves towards the water. Furthermore, all extended phases are found at elevated temperatures. Here, we find a 2-phase coexistence between different structures including polymeric micelles, droplet microemulsions (L_1_), bicontinuous microemulsions (µE), and lamellar microemulsions ($L_{\alpha }$). The detailed structures were resolved using SANS (Figure 10b). We could clearly identify lamellar structures by the first and second (or third) order peaks. Simply decaying functions identified droplets of large polydispersity. All of the discussed structures allow the system to circumvent strong curvatures towards the water, and so the droplets and micelles include oil as the inner phase. NMR measurements confirmed the preferential location of the side chains in water, which is the dominant aspect of counter-acting against the curvature at elevated temperatures.

### 4.3. Rationalization

The coefficients that describe the sensitivities of the comb polymer effect on the bending rigidity, the saddle splay modulus, and the mean curvature, namely, $\hat{\Xi }$, ϒ, and $\Xi_{\mathrm{SANS}}$, are plotted as a function of the side chain molar mass (Figure 11, Figure 12 and Figure 13). All coefficients were determined analogously to Comb 4 and the analysis is presented in Appendix A. As a guide of the eye, we have included linear regression lines for the polymer series, i.e., PEG, Comb 1, Comb 2, and Comb 3. All four polymers have nearly the same backbone fragment size with the different side chain lengths (Table 1). It can be seen that this series of data points already describes a well-defined trend for all three coefficients. Such a trend could be explained by the increasing effect of the side chain confinement along the backbone with increasing side chain length, which, in turn, influences the coefficients. For 2-dimensional brushes, a different scaling with the normalized grafting density is known [18]. However, this previously formulated that power law behavior does not necessarily apply in our case. Instead, it seems that in our representation, the three coefficients scale linearly with the side chain molar mass, *M*_SC_, which, in turn, would indicate an $M_{\mathrm{SC}}^{2}$ scaling for the bare elastic constants. Furthermore, the starting values for PEG are close to the coefficients of bare telechelic polymers [6] when the domain size (*d* = 103 Å = *d*_TS_/2) divided by the fragment end-to-end distance (∼30 Å) is approximately 2 ($\hat{\Xi }=-1$, ϒ=0 and $\Xi_{\mathrm{SANS}}=0$), i.e., close to weak confinement of the backbone. The comb polymers 2, 6, and 7 with differing backbone fragment length are nearly indistinguishable, whereas the comb polymers 1, 4, and 5 show a trend that points in the same direction as the growth of side chain length. In the latter case, the confinement is achieved by shortening the backbone fragment length. Why this is more pronounced than in the case of Combs 2, 6, and 7 is not immediately obvious.

Comb 8 strongly diverges from the observed trends, both in terms of magnitude and direction. For the mean curvature coefficient, ϒ, it can be argued that there is a considerable fraction of detached polymers that acts as polymer micelles or stars—due to the short backbone fragments—similar to a homopolymer [45]. This would also explain the large negative value found for $\Xi_{\mathrm{SANS}}$, which would be expected if the size of the micelles was on the order of the domain size. However, the presence of a large amount of detached polymer is not reflected in the $\hat{\Xi }$ value, which we would expect to be negative. It appears then that there is an equilibrium between anchored and detached polymer that yields these coefficients.

To rationalize the stretching of side chains and backbone fragments, we have developed a thermodynamic model that describes a stretched comb polymer along a planar membrane on the basis of entropic springs—similar to the elasticity of rubber [46]. We have already introduced a stretching factor for the backbone fragments, α. Assuming that the backbone anchoring side chains occupy elliptical spaces that touch each other (Figure 14), we obtain for the side chain stretching parameter:(26)$$\alpha_{\mathrm{SC}}=\alpha \frac{R_{\mathrm{eeBB}}}{R_{\mathrm{eeSC}}}$$

From the entropy of stretching [46], the backbone and the side chains, i.e., the rubber elastic entropy, we obtain the thermodynamic potential:(27)$$\frac{S}{k_{B}T}=\frac{\alpha^{2}R_{\mathrm{eeBB}}^{2}+2R_{\mathrm{eeBB}}^{2}\left(\frac{1}{\alpha }\right)}{R_{\mathrm{eeBB}}^{2}}+\frac{\alpha_{\mathrm{SC}}^{2}R_{\mathrm{eeSC}}^{2}+2R_{\mathrm{eeSC}}^{2}\left(\frac{1}{\alpha_{\mathrm{SC}}}\right)}{R_{\mathrm{eeSC}}^{2}}$$

The minimum entropy solution is found where:(28)$$\alpha =\sqrt[{3}]{\frac{1+1/r}{1+r^{2}}}\;\mathrm{and}\;r=\frac{R_{\mathrm{eeBB}}}{R_{\mathrm{eeSC}}}.$$

The calculated stretching parameters are given in Table 2. A clear trend of side chain stretching is observed for the series PEG, Combs 1, 2 and 3, which supports the trend observed in Figure 11, Figure 12 and Figure 13. The series Combs 2, 6, and 7 does not display considerably different values, which confirms highly similar values for $\hat{\Xi }$, ϒ, and $\Xi_{\mathrm{SANS}}$. The trend of Combs 1, 4, and 5 is not strongly pronounced. One could argue that the fragments are no longer Gaussian chains and so the confinement effects may appear more pronounced than for Gaussian chains. This topic still remains open for discussion.

When looking at the magnitudes of the polymer effect on the coefficients $\hat{\Xi }$, ϒ, and $\Xi_{\mathrm{SANS}}$, it can be seen that most of the comb polymers have a rather weak effect and may not be suitable for applications. These polymers are therefore predominantly only of interest for academical research. However, Comb 3 with longer side chains, and an intermediate backbone fragment length displays a strong boosting effect (i.e., $\hat{\Xi }$ ≈ 1.5) and a negative $\Xi_{\mathrm{SANS}}$. This is advantageous for the enhancement of the surfactant efficiency, i.e., reducing the amount of surfactant required for technical formulations. However, the negative $\Xi_{\mathrm{SANS}}$ also results in the coexistence of many liquid crystalline phases with the microemulsion without altering the original fish tail in the phase diagram. This moderate influence on the formation of liquid crystalline phases may be useful in applications where temperature switchable rheological properties are required, for instance. In any case, such comb polymers with longer side chains at low density may lead to interesting new effects for academic and applied research. However, the density of side chains must not be as high as for Comb 8 described by a value of *M*_BB_ = 280 g/mol.

## 5. Discussion

The results of the current study appear to agree well with results found for the series of polymeric architectures of amphiphilic polymers that we have studied so far [1,2,3,4,5,6,7,8,9]. Both hydrophobic and hydrophilic blocks of distinguished philicity were large enough that they would find their way to the preferred solvent in one domain of a bicontinuous microemulsion. This is important for the entropy-driven pressure that a polymer can exert on the membrane due to the configurational freedom while it is still anchored in the membrane. Historically, this effect was first established for diblock copolymers [1,2,3,4] where the symmetry of the system was maintained to a large extent. The so-called polymer-boosting effect describes the reduction in surfactant required to form a one-phase microemulsion. The flattened membrane with less fluctuations has a lower surface area to volume ratio, and so, the surfactant amount could be reduced. In the case of sticker polymers [5], the length of one block is reduced to a minimum, yet these blocks remained anchored in the membrane. The single-sided pressure is sufficient to induce the boosting effect although the preferred curvature of the membrane is influenced. Telechelic polymers [6] have two sticker units at either terminus of the polymer and therefore allow for bridging effects. Two neighboring points of the membrane could be linked by the polymer if the membranes are sufficiently diluted. If the concentration of the surfactant is higher, neighboring membranes can also be bridged. The essential parameter to distinguish these scenarios is the ratio between the domain size, *d* = *d*_TS_/2, and the end-to-end distance of the polymer, *R*_ee_. This concept of confinement could also be extended to diblock copolymers [8] (and homopolymers [9]) where the limited domain size presses the polymers against the membrane.

In this work, we have shown, using contrast variation SANS experiments, that the comb polymers considered here are also anchored to the membrane, and the different fragments of backbone and side chains find their way into the domain of preferred solubility. This leads to strong and well-defined anchoring of the fragments at the hydrophobic/hydrophilic boundary, which is not necessarily given [7]. The key point of our study is that we can now describe the effects of the comb polymer on the thermodynamics of the microemulsion by considering the comb polymer as a superposition of single backbone loops and side chains (all anchored in the membrane). The single fragments correspond directly to telechelic [6] and sticker [5] polymers. For the PEG with no side chains, the values start close to the ideal telechelic polymer values. Then with increasing side chain length, the coefficients increase in magnitude due to confinement effects. Such confinement effects were also observed in the series of Combs 1, 4, and 5. The concept of superposition had already been considered for the case of diblock copolymers and homopolymers [9], but had not yet been confirmed for more complicated architectures such as the combs, where the different fragments are covalently bound. Although we concede that the final coefficient plots (Figure 11, Figure 12 and Figure 13) leave some space for interpretation due to experimental uncertainties, the trends are clearly observable. Only Comb 8 does not perfectly anchor in the surfactant membrane and, therefore, behaves more like a water-soluble polymer due to micelle formation.

While in microemulsions the comb polymers unfold quite well and the fragments of different philicity are found in different domains, an important question is of the migration of substances through lipid bilayers and finally through cell walls. Balanced philicity was found to be the key for linear polymers [47], hinting at the possibility that drugs could be carried selectively into the cells by these new polymers. Comb or graft polymers can also serve as gene carriers [10,11,12]. A polyphosphoester has been shown to deliver DNA to muscle tissue [48]. It remains an open question how the polymer wraps DNA and what is the improved mechanism of this complex with membranes in the delivery.

## 6. Conclusions

We have investigated the structure of comb polymers in bicontinuous microemulsions. The polymers unfold and the fragments of different philicity are found in different domains. This leads to a strong and well-defined anchoring. The effect of the polymers on the microemulsion thermodynamics could be described by a superposition of the individual fragments (backbone and side chains) that correspond to telechelic and sticker polymers, respectively. In this way, the coefficients of the saddle splay modulus, the mean curvature, and the bending rigidity could be deconvoluted into contributions from bridging backbone fragments and single side chains. In many instances, the side chains exhibited brush-like characteristics due to the high grafting density. While the study found many examples with a rather weak effect on the microemulsion thermodynamics, Comb 3 with long side chains had a comparatively strong effect that was also found in many well-differentiated phases aside from the one-phase region that is usually the focus of such studies. This phase diagram supports the possibility of switching between different phases by slight temperature changes. So, the polymer could support dedicated applications where different behaviors of the microemulsion are required at different stages of a complicated process—in a similar manner to those found for telechelic polymers [6]. The whole study may also be considered in terms of balanced philicity, which becomes important for transmembrane migration of polymers alone or polymer–DNA complexes.

## Figures and Tables

**Figure 1 nanomaterials-10-02410-f001:**
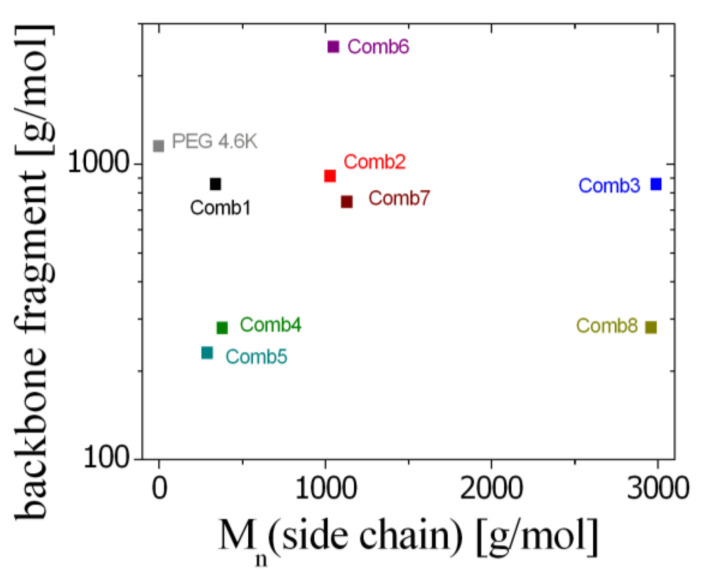
The molar masses of the backbone fragment (i.e., the average molar mass connecting two side chains, M_BBtot_/N_SC_) and the side chain for all comb polymers. A wide variety of parameters is investigated in this study.

**Figure 2 nanomaterials-10-02410-f002:**
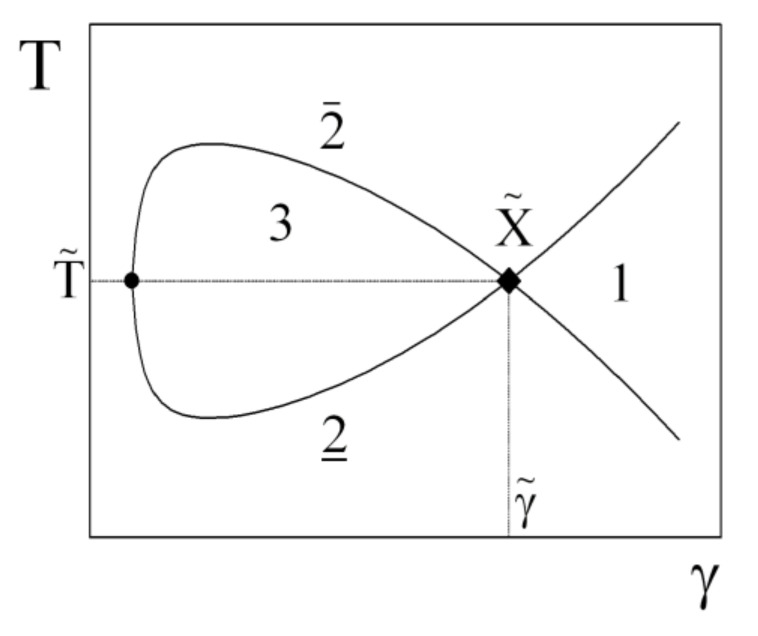
Scheme of a fish phase diagram of a microemulsion with nonionic surfactants: Temperature versus surfactant content. The different coexisting phases are indicated by the numbers. The fish tail point is indicated by $\tilde{X}=(\tilde{\gamma },\tilde{T})$

**Figure 3 nanomaterials-10-02410-f003:**
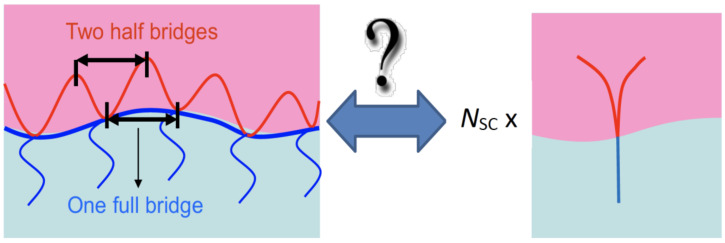
Sketch of the segmentation of a comb polymer to isolated side chains and half-loops. The backbone fragments as bridges are interpreted as half-segments.

**Figure 4 nanomaterials-10-02410-f004:**
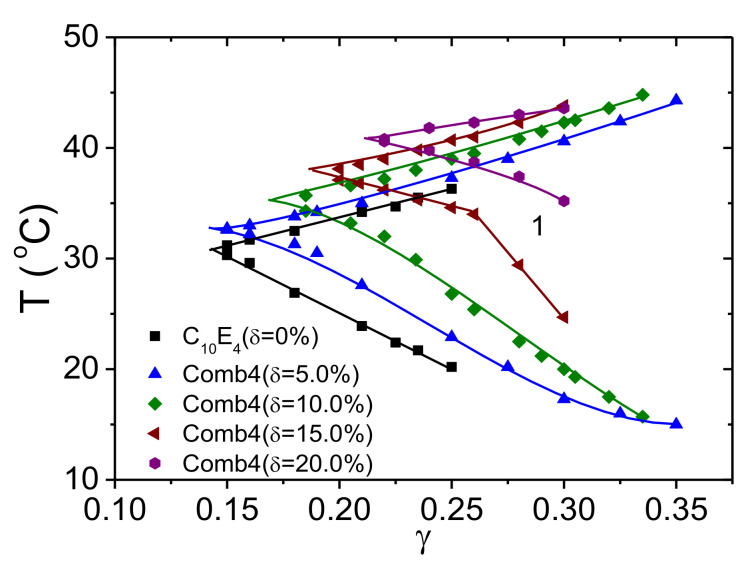
One phase regions of the phase diagrams (called fish tails): temperature versus surfactant content for the polymer Comb 4. With increasing polymer content, the required surfactant amount increases and the phase inversion temperature increases.

**Figure 5 nanomaterials-10-02410-f005:**
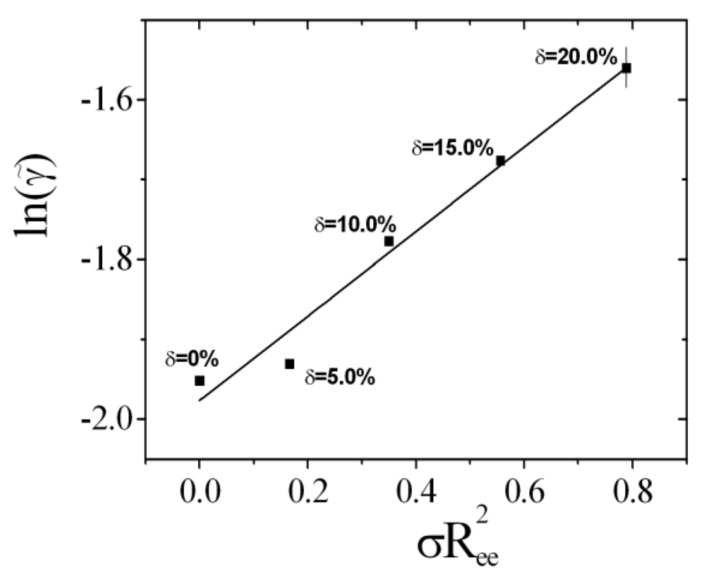
The logarithm of the minimum surfactant amount $\tilde{\gamma }$ for different fractions of Comb 4 as a function of the scaled polymer surface density $\sigma R_{\mathrm{ee}}^{2}$. The rightmost data point indicates the typical error bar.

**Figure 6 nanomaterials-10-02410-f006:**
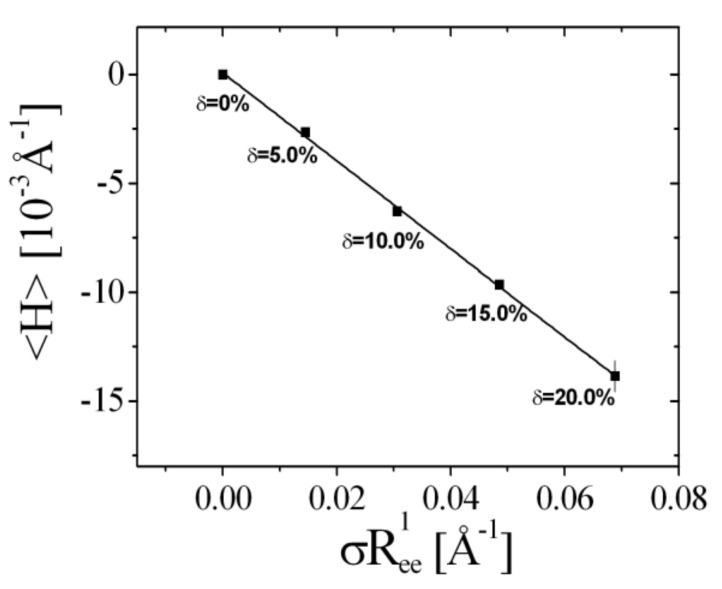
The mean curvature 〈H〉 obtained from the phase inversion temperature for different polymer amounts of Comb 4 as a function of the particular scaled polymer surface density, $\sigma R_{ee}^{1}$. The rightmost data point indicates the typical error bar.

**Figure 7 nanomaterials-10-02410-f007:**
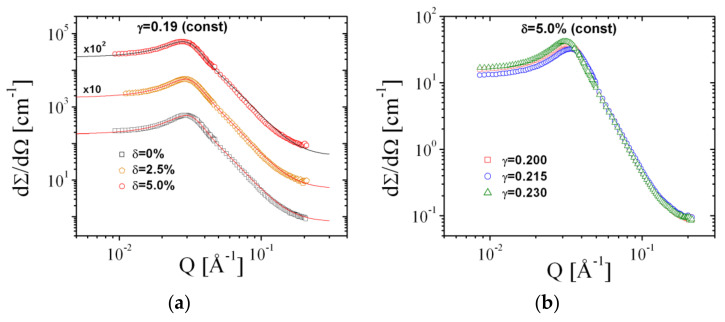
Small-angle neutron scattering (SANS) curves from Comb 4 with (**a**) different amounts of polymer and (**b**) with different amounts of surfactant. All curves are fitted with the Teubner–Strey model (Equation (11)). Statistical errors are within the symbol size.

**Figure 8 nanomaterials-10-02410-f008:**
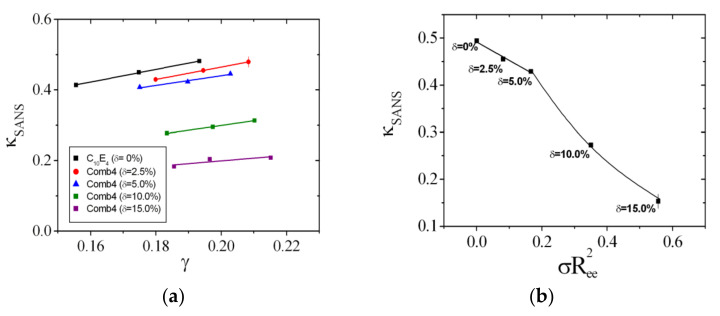
(**a**) The collection of bending rigidities from SANS experiments in units of *k*_B_*T*. The rightmost data point of the δ = 2.5% series indicates the typical error bar. (**b**) The dependence of the bending rigidity from SANS experiments on the scaled polymer amount. The slope can be used to determine the sensitivity $\Xi_{\mathrm{SANS}}$. The rightmost data point indicates the typical error bar.

**Figure 9 nanomaterials-10-02410-f009:**
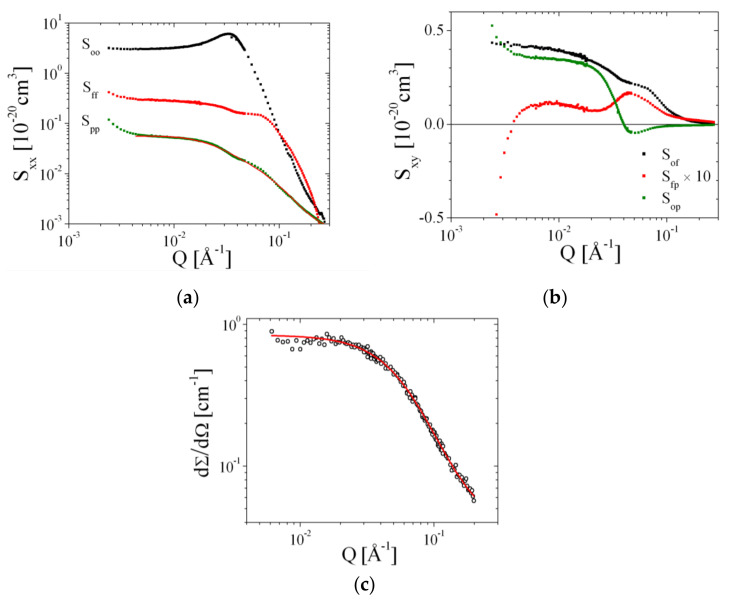
The deconvoluted scattering functions of the Comb 4 polymer in the microemulsion (**a**,**b**) and the polymer scattering in a good solvent (deuterated THF) (**c**). The polymer scattering is, in both cases, modeled as discussed in the main text.

**Figure 10 nanomaterials-10-02410-f010:**
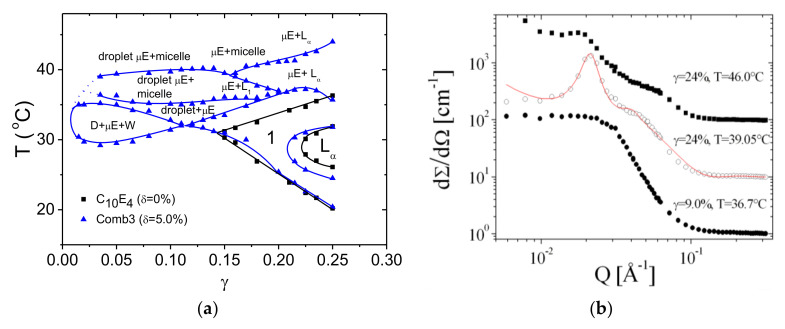
(**a**) Phase diagram showing temperature versus surfactant amount for the system containing Comb 3 (blue symbols and lines). The black symbols and lines show the behavior of the system without polymer. The new phases (coexisting phases) are denoted: (largely polymeric) micelles, droplet microemulsions (droplet µE, D, or L_1_), bicontinuous microemulsions (µE or 1), and lamellar microemulsions ($L_{\alpha }$). The expelled phases are either oil droplets (D) or water-rich phases (W). All the phases were supported by visual inspection and crossed polarizers, SANS measurements, and NMR. (**b**) Representative SANS patterns of the ordered phases to support the classification of the phase structure. The curves for 39.05 and 46.0 °C are multiplied by factors 10 and 100, respectively. A model fit for lamellar microemulsions is indicated by the red line.

**Figure 11 nanomaterials-10-02410-f011:**
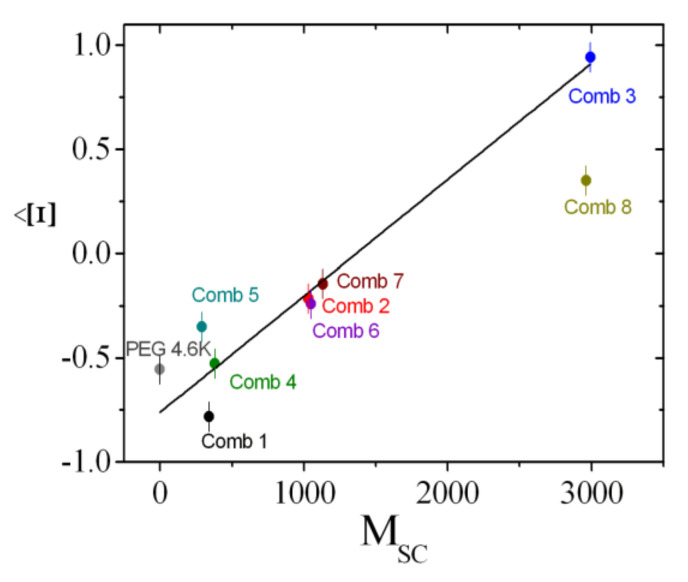
The sensitivity of the saddle splay modulus, $\hat{\Xi }$, as a function of the side chain molar mass, *M*_SC_. The values are determined from the phase diagram measurements (i.e., the minimum amount of surfactant, $\tilde{\gamma }$). The line is a guide for the eye that was fitted to the series: polyethylene oxide (PEG), Comb 1, 2, and 3.

**Figure 12 nanomaterials-10-02410-f012:**
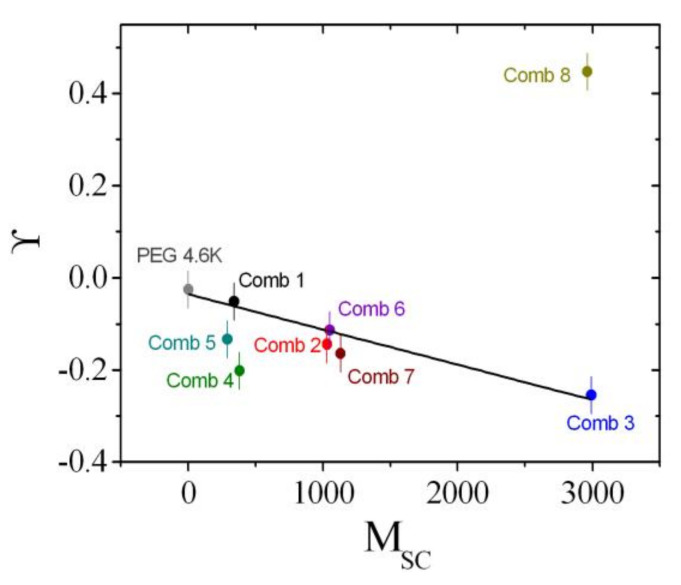
The sensitivity of the mean curvature, ϒ, as a function of the side chain molar mass, *M*_SC_. The values are determined from the phase diagram measurements (i.e., the phase inversion temperature, $\tilde{T}$). The line is a guide for the eye that was fitted to the series: PEG, Comb 1, 2, and 3.

**Figure 13 nanomaterials-10-02410-f013:**
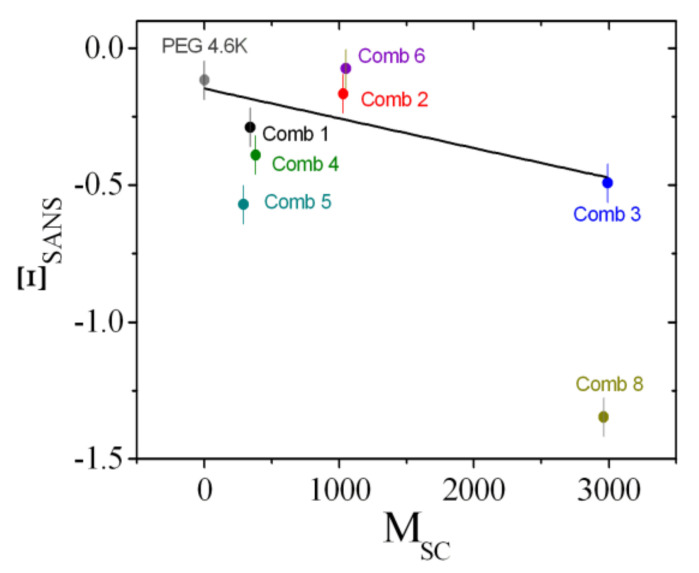
The sensitivity of the bending rigidity, $\Xi_{\mathrm{SANS}}$, as a function of the side chain molar mass, *M*_SC_. The values are determined from the small-angle neutron scattering measurements. We did not perform SANS measurements on Comb 7. The line is a guide for the eye that was fitted to the series: PEG, Comb 1, 2, and 3.

**Figure 14 nanomaterials-10-02410-f014:**
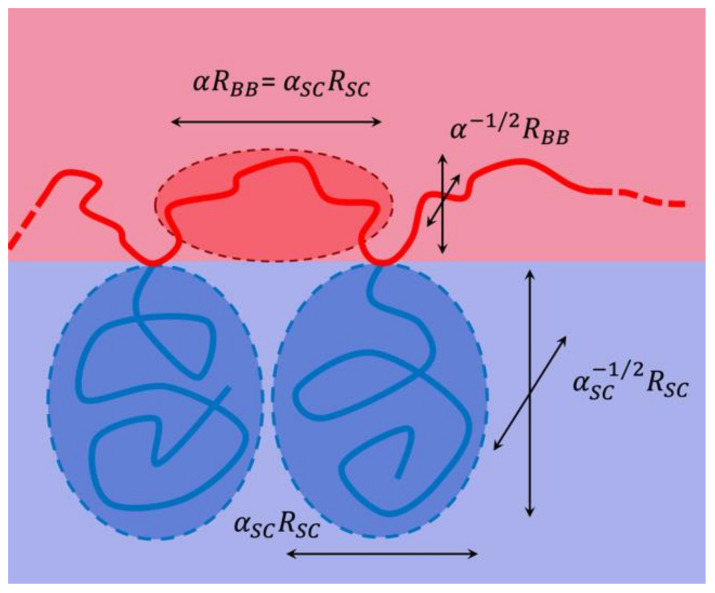
Sketch of the stretching model caused by the repelling side chains (blue) acting on the backbone (red). All of the polymers are in the microemulsion, either in the water (blue) or oil (red) domains.

**Table 1 nanomaterials-10-02410-t001:** Molecular weight characterization of the comb polymers.

	PBO Backbone		PBO-PEO Comb Polymer			
Polymer	M_BBtot_ ^a^ (g/mol)	PDI_BBtot_ ^b^	PDI_comb_ ^b^	m_SC_/m_BB_ ^c^	N_SC_ ^d^	M_SC_ ^e^ (g/mol)
Comb 1	11,200	1.03	1.04	0.40	13.1	340
Comb 2	11,200	1.02	1.05	1.13	12.3	1030
Comb 3	11,200	1.03	1.05	3.50	13.1	2990
Comb 4	10,300	1.03	1.07	1.36	36.9	380
Comb 5	11,400	1.03	1.07	1.14	44.8	290
Comb 6	29,000	1.07	1.07	0.42	11.6	1050
Comb 7	29,000	1.07	1.07	0.94	24.1	1130
Comb 8	10,300	1.03	1.03	10.49	36.5	2960
PEG 4.6 k	4600	1.05			(4.0)	

^a^ Number average molecular weight of the PBO backbone, obtained from SEC measurements; ^b^
*M*_w_/*M*_n_ (SEC); ^c^ mass ratio of PEO side chains to PBO backbone, obtained by ^1^H-NMR; ^d^ number of PEO side chains per PBO backbone, this value is identical to the number of OH-groups per backbone after the oxidation reaction (NMR); ^e^ number average molecular weight per side chain, calculated from *M*_BBtot_, *m*_SC_/*m*_BB_, and *N*_SC_.

**Table 2 nanomaterials-10-02410-t002:** The relaxed end-to-end distances and stretching parameters describing the confinement of the backbone and side chains. In the two rightmost columns, stretching is indicated by values larger than 1. Values smaller than 1 indicate no stretching and could be replaced by unity.

Polymer	M_BB_ ^a^ (g/mol)	R_eeBB_ (Å)	R_eeSC_ (Å)	α	$\alpha_{\mathrm{SC}}^{-1/2}$
Comb1	850	18.66	14.54	0.876	0.943
Comb2	910	19.35	27.66	1.177	1.102
Comb3	850	18.66	51.32	1.491	1.358
Comb4	280	9.75	15.51	1.229	1.138
Comb5	250	9.24	13.26	1.179	1.103
Comb6	2500	34.77	27.97	0.892	0.950
Comb7	1200	22.75	29.19	1.124	1.068
Comb8	280	9.81	51.02	1.815	1.693
PEG 4.6k	850	29.49	(0)	(1)	(1)

^a^ Mean backbone fragment molar mass.

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
