# Peer review of "Amphiphilic Comb Polymers as New Additives in Bicontinuous Microemulsions"

_nanomaterials, 2020, doi:10.3390/nano10122410_

Round 1

Reviewer 1 Report

This paper investigates structures of amphiphilic comb-like polymers consisting of hydrophilic sidechains and hydrophobic mainchain in microemulsions by using contrast variation SANS to discuss the effect of the amphiphilic comb-like polymer on the  microemulsions. 

It is significantly important and interesting result that the arrangement of each fragment of the comb-like polymers in the membrane is clarified by well-planned SANS experiments and detailed analysis. The discussion based on this result is rational and unquestionable. The reviewer considers this paper has significant value in both scientific and industrial fields, such as biomedical, paint, and so forth. Therefore, the reviewer considers this paper is worth to be published in Nanomaterials. Finally, the reviewer recommends that this paper undergoes an English proofreading.

Author Response

We would like to thank all referees for their constructive criticism. We hope that all of the criticism issues raised have been addressed is answered by our revision. All changes are marked in green. With these revisions we are confident that the manuscript is now suitable for publication. So we hope to get way for publication now.

We thank the reviewer for this positive reply. We have done a thorough proofreading. It was performed by a native speaker and we have decided to one redefined term: sidearm —> side chain. 

We kept the term “backbone fragment” as these are separate from and should not be confused with segment, such as the Kuhn segments.

Reviewer 2 Report

This paper regards with a study on the effects of amphiphilic comb polymers on the thermodynamics of bicontinuous microemulsions. Unfortunately the manuscript is poorly written; the introduction is very hard to understand, in particular it is not clear what the rationale is behind this study, including the hypotheses, the objectives, and the experimental design. I have also tried to read through the experimental section, however the Materials and Methods paragraph does not show the protocols and the technical details in a standard scientific format (for example, the experimental details of the polymer syntheses and the chemical characterisation are missing in the main text, and poorly described in the supporting info.). Under these circumstances, this paper does not reach the minimum scientific quality level for a publication in this journal.

I’m also confident that this work may present results which are potentially interesting for the scientific community, therefore I encourage the authors to rewrite their manuscript in a more comprehensive form and resubmit it.

Author Response

We would like to thank all referees for their constructive criticism. We hope that all of the criticism issues raised have been addressed is answered by our revision. All changes are marked in green. With these revisions we are confident that the manuscript is now suitable for publication.So we hope to get way for publication now.

We would like to thank the reviewer for raising these issues and can assure the reviewer that we have taken on board. The manuscript and SI have been performed a thorough English proofreading by a native speaker and we hope that the text is now easier to understand. We have added all SEC (GPC) traces and NMR spectra. So we therefore hope that this answers the all uncertainties that may have remained after we left open in the initial submission.

Reviewer 3 Report

  1. Saha et al. investigated the structure of comb polymers consisting of PBO-backbone and PEO-sidearms. The manuscript was well written and data are tolerable. I recommend that this article is acceptable for publication after revision. Following points should be revised before publication.

  1. The plots in Figure 1 are discrepancy with the values in Table 1. The y-axis value of Comb 1, that is the molecular weight of backbone, in Figure 1 is approximately 900 g/mol. However, the value of molecular weight of backbone is 11200. Other data are similar.
  2. The condition of SEC analysis including column name, elusion, flow rate and so on, are missing in the experimental section.
  3. Although the only SEC trace of Comb 8 is shown in the supporting information, all SEC traces and NMR spectra of synthesized polymer samples Comb 1-8 should be shown in the supporting information to confirm the data of polymer samples.

Author Response

We would like to thank all referees for their constructive criticism. We hope that all of the criticism issues raised have been addressed is answered by our revision. All changes are marked in green. With these revisions we are confident that the manuscript is now suitable for publication.So we hope to get way for publication now.

We thank the reviewer for raising these points. We have included a more detailed explanation of the y-axis of Fig. 1 means: A The mass of a backbone fragment is the average molar mass that is situated between two sequential side chains, i.e.so MBBtot/NSC. 

All SEC traces and NMR spectra have been added to the SI. We have also provided further details about the column, solvent and measurement conditions.